# Antioxidant Activity of *Vitis davidii* Foex Seed and Its Effects on Gut Microbiota during Colonic Fermentation after In Vitro Simulated Digestion

**DOI:** 10.3390/foods11172615

**Published:** 2022-08-29

**Authors:** Huiqin Ma, Aixiang Hou, Jiaojiao Tang, Aiai Zhong, Ke Li, Yu Xiao, Zongjun Li

**Affiliations:** 1College of Food Science and Technology, Hunan Agricultural University, Changsha 410128, China; 2Hunan Province Key Laboratory of Food Science and Biotechnology, Changsha 410128, China; 3Key Laboratory of Ministry of Education for Tea Science, College of Horticulture, Hunan Agricultural University, Changsha 410128, China

**Keywords:** *Vitis davidii* Foex seed, bioactive substances, antioxidants, in vitro digestion, in vitro fermentation, short-chain fatty acids

## Abstract

*Vitis davidii* Foex whole seed (VWS) is a by-product during the processing of grape products, which is rich in bioactive compounds that have great potential in the food industry. In this study, the bioactive compounds and antioxidant activity of VWS were determined, and their dynamic changes during in vitro colonic fermentation were also investigated after VWS subjected to in vitro simulated digestion. Results showed that VWS were rich in polyphenols (23.67 ± 0.52 mg GAE/g), flavonoids (13.13 ± 1.22 mg RE/g), and proanthocyanidins (8.36 ± 0.14 mg CE/g). It also had good DPPH and ABTS radical scavenging activity, which reached 82.10% and 76.10% at 1000 μg/mL. The alteration trend of the antioxidant activity during in vitro fermentation for 24 h was consistent with that of the content of bioactive substances, such as polyphenols, with the extension of fermentation time. The bioactive compounds and antioxidant activity showed a trend of increasing and then decreasing, reaching the highest value at 8 h. The high-throughput sequencing analysis of the regulatory effect of VWS on intestinal micro-organisms revealed that VWS influenced intestinal microbiota diversity. The relative abundance of beneficial microbiota, such as *Blautia* and *Parabacteroides*, increased by 4.1- and 1.65-fold after 24 h of fermentation compared with that of the control group. It also reduced *Escherichia-Shigella* by 11.23% and effectively reduced host inflammation, while increasing the contents of acetic acid, propionic acid, and other metabolites. Taken together, these results reveal the value of VWS utilization and provide new insights into the nutritional and microbiota modulation effects of VWS, which could therefore serve as a nutraceutical ingredient in health promotion.

## 1. Introduction

*Vitis davidii* Foex is a grape species belonging to Vitaceae, mainly distributed in Shaanxi Province, Gansu Province, and Central, Southern, and Southwestern China. *Vitis davidii* Foex has more juices and flavors than other grape varieties; for this reason, they are more suitable for juice processing, with a juice yield of up to 62%. They have many seeds that account for 4.2% of the total weight of fruits, which is higher than that in ordinary grapes (1.2% of the weight of the seeds) [1]. During winemaking, each kilogram of crushed grapes produces more than 0.2 kg of pomace, which is a major by-product of wineries [2]. However, nutrients in grape pomace, which are also rich in phenolic compounds, are not completely extracted during winemaking [3]. Grape seeds make up approximately 25% of pomace and are generally used as an animal feed additive, but it is not an optimal feed because of its low protein content and seasonal restrictions [4]. Moreover, most of the pomace is discarded, contributing to environmental pollution. Therefore, the rational use of grape seeds can avoid the wastage of resources and reduce environmental pollution.

Prebiotics are defined as substrates selectively utilized by the host’s micro-organisms resulting in benefits for metabolic health, the gastrointestinal system, and mental health [5]. Polyphenols are a group of the most extensive metabolites, which have phenolic structural features in nature [6], but because of the low bioavailability of nutrients, such as polyphenols, only a small fraction is directly absorbed by the small intestine, and up to 90% of these compounds are retained in the colon and metabolized by intestinal bacteria [7]. The role of polyphenols in health largely depends on their metabolism, absorption, and bioavailability processes, which are in turn related to the gut microbiota modulation in terms of composition and functionality [8]. Although polyphenols are currently considered to be modulators of gut microbiota composition, the prebiotic effect of each polyphenol may be influenced by the food source and chemical structure of the compound, as well as individual differences in the composition of the gut microbiota [9]. Gut micro-organisms are diverse and numerous; they can communicate not only amongst themselves to transmit information and substances but also with their hosts and can participate in host metabolic processes; thus, they regulate the conversion of important host substances and profoundly influence the host’s immune system and metabolism [10]. Age and body build affect intestinal microbiota, but diet is the main factor influencing changes in intestinal microbiota [11]. Moreover, in vitro models for gastrointestinal digestion and colonic fermentation are becoming more widely used because of the practical, economic, and ethical limitations of in vivo intervention trials [12,13]. For example, the SDS-III monogastric animal bionic digestive system is used to simulate human digestion. This dynamic system can better reflect actual digestion and reduce operational errors than traditional static simulated digestive systems [14]. In 2014, Minekus [15] proposed a generalized and standardized in vitro digestion method incorporating another proposed in vitro colonic fractionated fermentation model, which simulates human digestion and absorption; through this method, food metabolism and bioavailability can be comprehensively studied [16]. This in vitro model has been successfully applied to evaluate changes in the functional properties of chickpeas, oranges, tomatoes, and peanuts before and after in vitro fermentation; it has also been used to explore the effects of different foods on the structure of intestinal microbiota [17].

Few studies have investigated the interaction of various components of VWS with intestinal microbiota. Therefore, in this experiment, in vitro gastrointestinal digestion and colonic fermentation models were used to simulate the digestion and absorption of VWS in humans, explore the interaction between VWS and intestinal microbiota, and elucidate the potential effects of the components in VWS on intestinal health through their joint action. This study was also performed to evaluate their potential effects on human health and provide a reference for the utilization of wine by-products.

## 2. Materials and Methods

### 2.1. Material and Chemicals

The compounds 2, 2-diphenyl-1-picrylhydrazyl (DPPH), 2, 4, 6-tris (2-pyridyl)-S-triazine (TPTZ), 2, 2-azinobis (3-ethylbenzothiazoline-6-sulfonic acid) diammonium salt (ABTS), ascorbic acid (vitamin C, Vc), Gallic acid, rutin, catechins, Folin–Ciocalteu’s reagent were purchased from RYON Biotechnology Co. Ltd. (Shanghai, China), other chemicals and reagents were of analytical grade and purchased from Sinopharm Chemical Reagent Co. Ltd. (Shanghai, China).

*V. davidii* Foex seeds were taken from a wine factory in Hunan Province in China.

### 2.2. Sample Preparation and Extraction

The *V. davidii* Foex seeds were dried and crushed to 80-mesh powder. An appropriate amount of VWS was then weighed in an Erlenmeyer flask, and ethanol: water (40:60, *v*/*v*) at a ratio of 1:12 was added, then extracted with ultrasound at a frequency of 40 kHz at 45 °C for 55 min, and the filtrate was obtained by filtration, repeated five times. The obtained filtrate was concentrated with a rotary vacuum evaporator at 40 °C to eliminate the solvent, posteriorly freeze-dried, and stored at −20 °C until use.

### 2.3. Determination of Total Phenolics, Flavonoids and Oligomeric Proanthocyanidins Content

The lyophilized powder was dissolved in ethanol aqueous (60%, *v*/*v*) as a compound solution with a concentration of 1 mg/mL. Total phenolic content (TPC) was determined via the Folin–Ciocalteu method [18,19]. A standard curve was established by preparing a concentration gradient of gallic acid, and results were expressed as gallic acid equivalents per gram VWS (mg GE/g).

Total flavonoid content (TFC) was determined via the color development method using aluminum nitrate [20]. The standard curve was established by preparing a concentration gradient of rutin, and results were expressed as rutin equivalents per gram VWS (mg RE/g).

Oligomeric proanthocyanidin content (OPC) was determined via the vanillin-sulphuric acid method [21]. First, 2.5 mL of 1% vanillin-methanol solution and 2.5 mL of 32% concentrated sulphuric acid methanol solution were added to 1 mL of the compound solution, mixed well, and left at 32 °C for 20 min. Then, absorbance was measured at the wavelength of 500 nm. A catechin concentration gradient was prepared to establish a standard curve, and test results were expressed as catechin equivalents per gram VWS (mg CE/g).

### 2.4. Antioxidant Capacity Assay

The in vitro antioxidant capacity of the samples was determined using the method of Liu et al. [22,23], and grape seed extracts were prepared at concentrations of 200, 400, 600, 800, and 1000 μg/mL. Equal amounts of DPPH solution were added to the extracts and the scavenging activity of DPPH radicals was measured at 517 nm; ABTS stock solution was prepared and added to the samples in a 4:1 ratio to obtain ABTS radical cation scavenging activity at 734 nm; hydroxyl radical scavenging activity was measured by salicylic acid method; ferric reducing antioxidant power was measured by reducing Fe^3+^-TPTZ. Positive control was established by preparing ascorbic acid solution with the same concentration gradient.

### 2.5. In Vitro Digestion and In Vitro Colonic Fermentation

#### 2.5.1. In Vitro Digestion

The digestion of the oral cavity, stomach, and small intestine was simulated according to the optimized method of Brodkorb et al. [24]. Digestion was carried out in an oral-gastric-small intestine simulator with automatic enzyme addition and automatic cleaning to reduce systematic errors caused by manual operation.

First, the VWS were mixed with simulated saliva (75 U/mL salivary amylases) at 1:1 and shaken at pH 7 for 2 min. The oral digest was then diluted with simulated gastric fluid (2000 U/mL pepsin) at 1:1 (*v*/*v*), and digestion was continued at pH 3 for 2 h. The gastric digest was diluted with simulated intestinal fluid at 1:1 (*v*/*v*) and incubated with bile salts and trypsin (at 100 U/mL of trypsin) at pH 7 for 2 h. At the end of in vitro digestion, the supernatant and precipitate of the digested samples were separated. The precipitate was lyophilised and stored separately from the supernatant in a refrigerator at −20 °C to obtain the test group digest. Because it is known that (on average) 10% of the supposedly absorbable fraction in the large intestine is actually not absorbed, 10% of all the fermentation supernatant is taken and mixed with all the precipitated lyophilised material to obtain a mixed sub-strate for in vitro fermentation [23]. Under the same conditions, digests without VWS were prepared as blank group digests (Figure 1). 

#### 2.5.2. In Vitro Colonic Fermentation

In vitro colonic fermentation was simulated in accordance with previously described methods [25] with slight modifications. Fresh whole stools were collected from subjects (two females and one male, who had not taken antibiotics within 3 months before collection of fecal samples), processed, and fermented in an anaerobic incubator.

A sufficient amount of basal growth medium was prepared (peptone water 2 g/L, yeast extract 1 g/L, NaCl 0.1 g/L, K_2_HPO_4_ 0.04 g/L, KH_2_PO_4_ 0.04 g/L, MgSO_4_⋅7H_2_O 0.01 g/L, CaCl_2_⋅2H_2_O 0.01 g/L, NaHCO_3_ 2 g/L, bile salts 0.5 g/L, L-cysteine hydrochloride 0.5 g/L, hemin 50 mg/L, vitamin K1 10 μL/L, and Tween 80 2 mL/L). The fecal samples of the three volunteers were mixed equally, weighed, vortexed, shaken in sterile PBS buffer for 3 min, mixed again, and filtered through four gauze layers to obtain a 10% fecal suspension. Next, 100 mL of PBS mixture was added to a reagent bottle containing 900 mL of sterile nitrogen-containing basal medium. After the resulting medium was vortex-shaken for 3 min and mixed, the mixed medium containing the intestinal microbiota was obtained. The medium was divided into two portions; one was added with 5 mL of the digest of the blank group, and the other was added with 5 mL of digest mixture prepared in Section 2.5.1 and vortex-shaken for 3 min. After they were thoroughly mixed, the fermentation samples of the control group (group C) and the test group (group GS) were obtained. Groups C and GS were divided into 50 tubes of 10 mL/tube under anaerobic conditions and placed in an incubator at a constant 37 °C for static anaerobic fermentation. Subsequently, 10 tubes of fermentation broth were taken and stored at −80 °C at 0.5, 4, 8, 12, and 24 h of incubation, for the next experiments.

In vitro fermentation samples of 0.5, 4, 8, 12, and 24 h were taken from groups C and GS. For group C, these samples were labeled as control 0.5, control 4, control 8, control 12, and control 24 (C0.5, C4, C8, C12, and C24, respectively). For group GS, they were named grape seed 0.5, grape Seed 4, grape seed 8, grape seed 12, and grape seed 24 (GS0.5, GS4, GS8, GS12, and GS24, respectively). Thus, the samples were prepared to determine their TPC, TFC, OPC, and antioxidant properties via the same method as outlined in Section 2.3 and Section 2.4.

### 2.6. DNA Extraction and Sequencing

The genomic DNA of the microbial community was extracted from samples by using an EZNA^®^ Soil DNA Kit (Omega Bio-Tek, Inc., Norcross, GA, USA) in accordance with the manufacturer’s instructions. The quality of the DNA extract was checked on 1% agarose gel, and DNA concentration and purity were determined using a NanoDrop 2000 UV–vis spectrophotometer (Thermo Fisher Scientific, Wilmington, DE, USA).

The V3–V4 hypervariable region of the bacterial 16S rRNA gene were amplified with the primer pairs 338F (5ʹ-ACTCCTACGGGAGGCAGCAG-3ʹ) and 806R (5ʹ-GGACTACHVGGGTWTCTAAT-3ʹ) and 1737F(5ʹGGAAGTAAAAGTCGTAACAAGG-3ʹ) and 2043R (5ʹ-GCTGCGTTCTTCATCGATGC-3ʹ) by using an ABI Gene Amp^®^ 9700 PCR thermos cycler (Applied Biosystems, Foster City, CA, USA). The PCR amplification was performed as follows: initial denaturation at 95 °C for 3 min, followed by 27 cycles of denaturation at 95 °C for 30 s, annealing at 55 °C for 30 s, extension at 72 °C for 45 s, single extension at 72 °C for 10 min, and final incubation at 4 °C. The PCR mixtures were composed of 4 μL of 5 × TransStart FastPfu buffer, 2 μL of 2.5 mM dNTPs, 5 μM of each primer (0.8 μL), 0.4 μL of TransStart FastPfu DNA Polymerase, 10 ng of template DNA, and 20 μL of ddH_2_O. PCR was performed in triplicate, and PCR products were extracted from 2% agarose gel, purified using an AxyPrep DNA Gel Extraction Kit (Axygen Biosciences, Union City, CA, USA) in accordance with the manufacturer’s instructions and quantified using a Quantus™ fluorometer (Promega, Madison, WI, USA).

The purified PCR products were pooled at equimolar ratios and paired-end sequenced on an Illumina MiSeq PE300 platform/NovaSeq PE250 platform (Illumina, San Diego, CA, USA) in accordance with the standard protocols supplied by Majorbio Bio-Pharm Technology Co., Ltd. (Shanghai, China). Raw gene sequencing reads were demultiplexed, quality-filtered by fastp version 0.20.0, and merged using FLASH version 1.2.7. Operational taxonomic units (OTUs) with a 97% similarity cutoff were clustered using UPARSE version 7.1, and chimeric sequences were identified and removed. The taxonomy of each OTU representative sequence was analyzed using RDP Classifier (version 2.2) against the 16S rRNA database (Silva v138) at a confidence threshold of 0.7.

### 2.7. Analysis of pH Values and SCFAs Production

The fermentation broth of groups C and GS removed at 0.5, 4, 8, 12, and 24 h of fermentation and its pH was measured with a pH meter.

The SCFAs content of the samples at different fermentation times was determined via gas chromatography-mass spectrometry (GC-MS). The supernatant of colonic fermentation was mixed with 25% metaphosphoric acid solution at 4:1, vortexed for 2 min, centrifuged at 10,000 rpm for 5 min and filtered through a 0.22 μm microporous membrane.

The chromatographic conditions were as follows: chromatographic column, DB-FFAP gas chromatographic column (30 m × 250 μm × 5 μm); carrier gas, 99.99% high-purity nitrogen at a flow rate of 0.8 mL/min; and auxiliary gas, 99.99% high-purity hydrogen. The FID detector temperature, inlet temperature, splitting ratio, and injection volume were 280 °C, 250 °C, 5:1, and 1 μL, respectively. For the programmed temperature increased, initial temperature was set at 60 °C. Then, this temperature was increased to 220 °C at a rate of 20 °C/min and maintained for 1 min. Mass spectrometry conditions were set as follows: voltage of 70 eV, ion source temperature of 230 °C, mass scan range of *m*/*z* 28–300, and electron ionization (EI) mode. The mass spectra of the compounds were compared with the NIST17.L mass spectrometry database, and compounds with a match greater than 80% were extracted for analysis.

### 2.8. Statistical Analysis

Experimental data were the means of three replicates, and results were expressed as mean ± standard deviation. SPSS 26.0 software was used for statistical analysis, Duncan’s method was used for multiple comparisons, and Origin 2018 and GraphPad Prism 6.01 were used for graphing.

## 3. Results and Discussion

### 3.1. TPC, TFC, OPC, and Antioxidant Activity of VWS

The TPC, TFC, and OPC in VWS were 23.67 ± 0.52 mg GAE/g, 13.13 ± 1.22 mg RE/g, and 8.36 ± 0.14 mg CE/g, respectively (Figure 2). The polyphenol content in grape seeds varies widely from 12 mg/g–113 mg/g amongst grape seed varieties [26,27]. Song [28] also analyzed the TPC of different varieties of grape seeds and found that European grapes have the highest polyphenol content (103 mg/g), whereas *V. davidii* Foex and *Vitis quinquangularis* Rehd have the lowest polyphenol contents (37.1 and 35.6 mg/g, respectively). Makris extracted proanthocyanidins from wine-produced grape seeds and found that the OPC of fresh and wine-produced grapes are 27.0–43.3 and <10 mg/g, respectively. Similar to OPC, TPC and TFC are also reduced by winemaking [29]. In the present study, because the samples were obtained from *V. davidii* Foex winery by-products, the TPC, TFC, and OPC were lower than reported in the literature.

The antioxidant activity of VWS is shown in Figure 3A–D. VWS extract showed a good scavenging ability of DPPH radicals, reaching more than 50% at a sample concentration of 400 μg/mL. When the concentration was 800 μg/mL, the DPPH radicals scavenging ability of VWS was 80.70 ± 0.35%, which was similar to Vc at the same concentration (88.64 ± 1.53%). This change gradually leveled off, and the scavenging ability was no longer enhanced. In Figure 3B, the scavenging power of VWS for ABTS radical cation increased as the mass concentration increased and showed a good linear relationship. The scavenging rate was 76.10 ± 0.57% when the concentration was 1000 μg/mL, and the scavenging power of ABTS radical cation likely increased by increasing the concentration of the sample. The trend of the reducing power of ferrous ions is presented in Figure 3C. In the test concentration range, VWS had a weak reducing power of up to 266.19 ± 5.37 μmol/L only for ferrous ions, and the reducing power of Vc at the same concentration was six times higher than 266.19 ± 5.37 μmol/L. In Figure 3D, the VWS extract did not elicit a strong scavenging effect on hydroxyl radicals in the tested concentration range. The scavenging ability of hydroxyl radicals was also 14.27 ± 1.15% only at the maximum sample concentration of 1000 μg/mL. Some studies have shown that the scavenging ability of hydroxyl radicals depends on polyphenol type rather than its content [30]. Gülçin et al. [31] found that the position and number of hydroxyl and carbonyl groups in a polyphenol molecule affect its iron-chelating ability; furthermore, addition of hydroxyl and carbonyl groups to the 3′-, 4′-, and 5′-positions of the B-ring enhances its antioxidant capacity compared with that of single hydroxyl groups. Yildirim et al. [32] found that most of the phenolic compounds found in grapes can act as antioxidants and that the contents of catechins, epicatechin, and proanthocyanidin B2 have a highly significant positive correlation with in vitro antioxidant capacity (*p* < 0.01) [33]. Lv et al. [34] reported the antioxidant activities of 32 varieties of litchi and found that the higher the OPC, the stronger the free radical scavenging activity; therefore, the mechanism of antioxidants in litchi may involve free radical scavenging by catechins, epicatechin, and proanthocyanidins, the aggregates formed by catechin and epicatechin. Moreover, flavonoids have a significant free radical scavenging ability and can exert antioxidant activity by chelating variable metal ions [35]. These findings suggested that the antioxidant capacity of VWS is high possible because they are rich in phenolics, flavonoids, and proanthocyanidins.

Although grape seeds, as a by-product of winemaking, are not as rich in bioactive substances as fresh grape seeds, their TPC and TFC are in the middle to upper range compared with fruits, such as popcorn, wolfberry, and banana [36,37]. Babbar [38] compared the TPC of six fruit by-products and found that grape seeds contain 37.4 mg GAE/g of total phenols, which is 10 times more than in citrus seeds and banana peels. Overall, VWS are simple to obtain as an oenological by-product of wine, and their functional activity is not inferior to that of some fresh fruits. Moreover, their bioavailability and true benefits in vivo can be further determined through in vitro and in vivo tests.

### 3.2. Effect of In Vitro Fecal Fermentation on the Dynamic Change of TPC, TFC, and OPC

Phytochemicals undergo great alteration during gastrointestinal digestion. To exert their health effects, the bioactive compounds of VWS must be available in target tissues; thus, the biological activities of VWS may depend on their absorption and bioavailability in the intestinal tract. Thus, the effects of in vitro colonic fermentation on bioactive compounds and the antioxidant capacity of digested VWS were investigated. During in vitro fermentation, the TPC, TFC, and OPC initially decreased and then increased with the extension of fermentation time; after 8 h of in vitro fermentation, their highest values were obtained and showed a 45.45%, 42.61%, and 165.51% increase compared with those at the initial stage, respectively (Table 1). Then, these contents gradually decreased. During the fermentation stage for 0.5–8 h, TPC, TFC, and OPC showed an overall increasing trend probably because covalent, hydrogen, and hydrophobic bonds between the bound polyphenols and cell wall components in VWS are broken due to microbial action during microbial fermentation and released in their free states; as a result, these contents significantly increase [39]. At 8 h of colonic fermentation, all polyphenols were released and possibly decomposed by colonic micro-organisms as the fermentation time was extended; consequently, their content decreased. When the fermentation time reached 24 h, all the active substances decreased and reached the lowest value. Research by de Almeid [40] found that the free phenol content initially increases significantly (*p* < 0.05) after 4 h of fermentation and decreases significantly (*p* < 0.05) after 24 h of fermentation compared with those in the pre-fermentation period. Gowd [41] investigated the effect of in vitro fermentation on waxberry and found that the TPC and TFC initially decrease with the extension of fermentation time, subsequently increase to their maximum values, and gradually decrease; the antioxidant capacities of the samples changed with the active substances and decreased to the lowest values after 24 h of fermentation, similar to the results of this experimental study. Furthermore, the TPC of plant extracts from different sources gradually decreases after 24 h of in vitro fermentation [42].

### 3.3. Effect of In Vitro Fermentation on the Change of Antioxidant Activity

The activity of each antioxidant changed significantly during 24 h of in vitro fermentation (Figure 4). In particular, the activity of DPPH was much higher than the rest of the indicators. The result was consistent with the antioxidant capacity of VWS when they were undigested and fermented. At 8 h of fermentation, the DPPH radical scavenging power, ABTS radical cation scavenging power, ferrous ion reducing power, and hydroxyl radical scavenging power of the samples peaked at 12.68 mg Vc/mL, 0.50 mg Vc/mL, 1.96 mmol Fe^2+^/mL, and 0.47 mg Vc/mL. Moreover, the antioxidant capacity of the samples decreased and gradually stabilized at 12 h of fermentation. Combined with Table 1, TPC, TFC, and OPC peaked at 8 h of colonic fermentation, indicating that these active ingredients were related to antioxidant activity as the phenolic compounds in the samples started to be released under the action of colonic micro-organisms at the early stage of fermentation; then, the antioxidant capacity increased. With the extension of fermentation time, the polyphenolic substances were decomposed or transformed into small molecule metabolites, leading to a decrease in antioxidant activity. Correa [43] investigated Merlot grape seeds subjected to gastrointestinal digestion and in vitro fermentation and revealed that the antioxidant function of grape seeds and the bioactivity of nutrients, such as polyphenols are closely correlated at different fermentation stages. Del Pino-Garcia [44] confirmed this result and found red wine pomace positively affects the total antioxidant capacity after in vitro gastrointestinal digestion and colonic fermentation. Other studies have also shown that polyphenols and flavonoids are strongly correlated with antioxidant activity [45,46], which is consistent with the pattern of variation identified in the present study.

### 3.4. Effect of VWS Digest on Intestinal Microbiota

Human health is closely related to intestinal microbiota, and grapes are rich in active ingredients that can maintain intestinal health by regulating the intestinal microbiota [47]. Related studies have shown that grape seeds are rich in active substances that promote the growth of potentially beneficial bacteria, reduce the number of undesirable bacteria, such as *Clostridium perfringens*, and improve the intestinal environment; consequently, SCFAs, metabolites of beneficial bacteria increase, intestinal pH decreases, and human intestinal health improves [48]. To further analyze the effect of VWS on host intestinal microbiota, we determined the microbial composition of fecal micro-organisms at different fermentation sites by the high-throughput sequencing of bacterial 16S rRNA genes.

The diversity indices of the 10 samples were different (Table 2). The comparison of the changes pattern of the Chao1 index between groups C and GS at the same time showed that the Chao1 index of group C was higher than group GS at 8 and 12 h of fermentation. Thereafter, the Chao1 index of group GS was higher than group C, and the highest Chao1 index amongst all fermentation samples was observed in GS4. These findings indicated that VWS helped to improve the Chao1 index and microbial richness. The changes in the pattern of the ACE index were consistent with the Chao1 index; that is, both were higher in group C than in group GS at 8–12 h, and the highest value of 745.20 in group GS was higher than 622.45 in group C. The Shannon and Simpson indices represent microbial diversity, and the higher the value, the greater the microbial diversity. In this study, the Shannon and Simpson indices were lower in group GS than in group C during the pre-fermentation period probably because of the antibacterial effect of VWS; some studies have shown that the addition of grape polyphenols has an overall antibacterial effect on the intestinal microbiota, but this effect gradually disappears with time [48]. After 24 h of fermentation, the Shannon and Simpson indices of group GS were 3.035 and 0.618, respectively, which were higher than group C. Consistent with the finding of Cueva et al. [48], our results indicated that VWS were beneficial to microbial diversity.

The comparison of the overall data of the two groups, revealed that the abundance and diversity of microbiota in each group decreased after 24 h of fermentation compared with the initial stage, but each index was higher in group GS than in group C at 24 h. These findings indicated that VWS increased the richness and diversity of microbiota in the intestine. However, this decrease in richness and diversity can lead to intestinal microbiota dysbiosis, which can cause low inflammation and metabolic diseases; conversely, a rich and diverse intestinal microbiota is more beneficial to health [49].

Evolutionary clustering was analyzed in terms of LDA Effect Size (LEfSe) to identify statistically significant dominant micro-organisms. Figure 5 showed that the bacterial groups in groups C and GS differed after 24 h of in vitro fermentation; some clusters increased, whereas others decreased. At the phylum level, the relative abundance of Firmicutes and Bacteroidetes was significantly higher in group GS24 than in group C24, conversely, the relative abundance of Proteobacteria was significantly lower. At the genus level, *Parabacteroides* was more abundant in group GS than in group C, whilst *Escherichia-Shigella* was more abundant in the group C than in group GS.

Figure 6A shows the gates with relative abundance greater than 1%. At 0.5 h, Firmicutes, Proteobacteria, Bacteroidetes, and Actinobacteria were the four major clades, with Firmicutes and Proteobacteria accounting for more than 90% of the total sequence reads, consistent with the results of related studies [50]. Figure 6C shows that after 24 h of in vitro fermentation, groups C and GS differed significantly in the gate level for Firmicutes and Proteobacteria. Firmicutes dominated at the early fermentation stage, and its abundance gradually decreased with time. After 24 h of fermentation, only 5.19% and 16.05% relative abundance remained in groups C and GS, respectively, and the value in group GS was three times higher than in group C. Most of the beneficial bacteria in the intestinal microbiota belonged to Firmicutes [51]. The correlation heat map (Figure 7) showed that the genera, such as *Butyricicoccus*, *Faecalibacterium*, *Blautia*, and *Roseburia*, belonging to Firmicutes were positively correlated with TPC and OPC, indicating that grapes have a promotional effect on the growth and reproduction of these beneficial bacteria. Gil-Sánchez [52] studied the effect of grape seed polyphenols on human intestinal microbiota and found an increase in the main bacteria during fermentation compared with that in the control group; these findings were consistent with the correlation analysis of the present study. During fermentation, the abundance of Proteobacteria increased and then decreased in groups C and GS; specifically, it increased to 83.59% at 8 h and then decreased to 79.28% at 24 h in group C. Group GS showed the same trend as group C but decreased to 64.28% at 24 h, which was slightly lower than the abundance in group C. TPC and OPC were negatively correlated with Proteobacteria and Bacteroidetes, indicating that the active substances in VWS inhibited the growth of Proteobacteria and Bacteroidetes. The increased abundance of Proteobacteria reflects the unstable structure of the intestinal microbial community and serves as a potential diagnostic criterion for diseases that may trigger inflammation [53]. Aura [54] also found that polyphenols elicit a significant inhibitory effect on the growth of Proteobacteria, and the intake of moderate amounts of polyphenols may reduce the risk of colon cancer. In the present study, the abundance of Proteobacteria in group GS was significantly lower than in group C. Therefore, VWS could reduce the increase in the abundance of Proteobacteria and alleviate host intestinal inflammation.

Figure 6B shows the distribution of the relative abundance greater than 1% at the genus level. Specifically, the intestinal microbiota was mainly composed of *Escherichia-Shigella*, *Faecalibacterium*, *Romboutsia*, *Blautia*, and other genera. Amongst them, *Escherichia-Shigella* dominated in the middle and late fermentation stages, peaked at 82.49% in group C8, and decreased to 72.28% after 24 h. Its abundance in group GS was significantly lower than in group C (Figure 6D). After 24 h of in vitro fermentation, its relative abundance in group GS was 61.05%, which was 11.23% lower than in group C. Pasqua [55] found that plant polyphenols can increase the permeability of bacterial cell membranes, causing intracellular ATP efflux and consequently acting as an antibacterial agent. Figure 7 TPC and OPC are negatively correlated with *Escherichia-Shigella*, indicating that grape seeds inhibit the growth of *Escherichia-Shigella* and attenuate inflammatory responses, ulcers, and hemorrhagic or mucous diarrhea [56]. The potential prebiotic effect of VWS on fecal bacteria was mainly reflected in the promotion of the growth of *Faecalibacterium* [57], in the present study, its percentage was low in all fermentation stages in group C, and its maximum value was only 8.32%; however, this percentage significantly increased in group GS, and the maximum value was 29.98%. In addition to *Escherichia-Shigella* and *Faecalibacterium*, *Romboutsia*, *Blautia*, and *Parabacteroides* exhibited more obvious variations. *Romboutsia* in both groups C and GS decreased, i.e., from 13.43% at 0.5 h to 1.22% at 24 h in group C. In group GS, the percentage decreased from 16.67% to 2.90% after 24 h of fermentation; this trend in group GS was almost similar to group C but with a slightly higher overall content. *Blautia* and Parabacteroides were effective weight-loss bacteria [58]; after 24 h of fermentation, their relative abundance in group GS increased by 4.1- and 1.65-fold, respectively, compared with group C. These results suggested that VWS might play a role in obesity symptoms. In summary, VWS have a more obvious effect on the growth of beneficial bacteria in the intestinal tract, and their active substances, such as polyphenols and proanthocyanidins, can regulate the balance of intestinal microbiota and maintain colon health.

### 3.5. Effect of VWS Digest on pH and SCFAs Production during In Vitro Fermentation

Acidic substances, such as SCFAs, produced through the fermentation of human intestinal micro-organisms decrease intestinal pH; therefore, the fermentation of intestinal micro-organisms and changes in their components can be indirectly reflected by changes in intestinal pH [17]. In Figure 8, during 24 h of in vitro fermentation, the pH in groups GS and C decreased and then increased; at 12 h, the lowest pH was obtained. The decrease in the pH of the fermentation broth was mainly the result of acid production from substrate fermentation; during fermentation, intestinal micro-organisms grew rapidly in the first 12 h and accumulated a large number of metabolites, which caused the decrease in pH. Conversely, the nutrients were gradually consumed at the later fermentation stage, and microbial growth was restricted; conversely, pH rebounded at the later stage, but the changes did not fluctuate greatly. Human intestinal pH in the range of 6–7 is beneficial to health [59]. The pH in group GS significantly decreased at all stages compared with that in group C, probably because the nutrients in VWS promoted the growth of intestinal micro-organisms and increased acid production from metabolism; thus, pH was affected. Consistent with the present results, previous findings on extracted ginger polyphenols for in vitro fermentation showed that pH decreases as fermentation time is prolonged [60]. Active ingredients, such as polyphenols and flavonoids, undergo microbial fermentation in the intestine to produce SCFAs, such as acetic acid and propionic acid, thus decreasing pH [61].

The main metabolites of carbohydrates fermented by intestinal micro-organisms during in vitro fermentation are SCFAs, whose production is related to the abundance and structure of intestinal microbiota [62]. SCFAs mainly include formic acid, acetic acid, propionic acid, butyric acid, isovaleric acid, and valeric acid; amongst them, the most abundant are acetic acid, propionic acid, and butyric acid, which can effectively inhibit the production of pro-inflammatory cytokines, enhance mucin expression, and maintain the immune function of the intestinal barrier [63]. At each fermentation stage, the contents of formic acid, acetic acid, propionic acid, and butyric acid increased substantially in group GS compared with those in group C (Figure 9); after 24 h of fermentation, SCFAs in group GS were 78.33%, 14.16%, 19.2%, and 139.31% higher than those in group C, respectively. Figure 9A–C demonstrate that the contents of formic acid, acetic acid, and propionic acid in the fermentation broth were low at 0.5–4 h of fermentation; at this time point, formic acid was not detected because of its low content. With the extension of fermentation time, formic acid, acetic acid, and propionic acid were produced in large quantities in each group, and their contents increased significantly, which was consistent with the trend shown by pH at different times. Butyric acid concentration gradually decreased (Figure 9D). Butyric acid is mainly produced by thick-walled bacteria, *Faecalibacterium* and *Romboutsia* detected in this study are typical butyric acid-producing bacteria [64]; therefore, a decrease in butyric acid content may be related to the substantial decrease in the relative abundance of such bacteria at the late fermentation stage. Amongst the individual SCFAs produced, acetic acid had the highest content, followed by propionic acid, which was closely related to certain strains producing SCFAs; carbohydrate-metabolizing bacteria, such as Bacteroidetes, produce acetic acid, which is consistent with the findings of Wang et al. [60]. Furthermore, polyphenols can induce an increase in SCFAs [65]. Martín [66] fed grape pomace to rats and found that it produces higher levels of acetic acid, propionic acid, and butyric acid than in the control group after 24 h. The growth and reproduction of numerous beneficial bacteria largely increase the concentration of SCFAs, which contribute to the acidification of the in vitro fermentation environment; this observation is also consistent with the changes in pH. In turn, the acidification of the colonic environment can promote the proliferation of beneficial bacteria and inhibit the growth of pathogenic bacteria. These processes play a key role in VWS-induced improvement of human intestinal health.

## 4. Conclusions

In this study, the effects of in vitro colonic fermentation on bioactive compounds and the antioxidant capacity of digested VWS were investigated. Furthermore, the effect of digested VWS on modulation of microbial community structure and SCFAs content were also studied. The results showed that the TPC, TFC, OPC, and antioxidant activity of in vitro digested VWS changed greatly during the colonic fermentation process. At 8 h, the TPC, TFC, and OPC increased by 45.45%, 42.61%, and 165.51%, respectively, compared with those at the initial fermentation stage. Thereafter, they gradually decreased as the fermentation progressed. The change of antioxidant activity is in accordance with the bioactive compounds. Furthermore, VWS have great effect on intestinal micro-organisms and modulates the structure of microbiota. The abundance of Escherichia-Shigella in group GS decreased by 11.23% compared with group C after 24 h of fermentation. VWS inhibited the growth and reproduction of harmful microbiota and increased the relative abundance of beneficial bacteria, such as *Faecalibacterium*, *Romboutsia*, and *Blautia*, to further adjust intestinal pH and increase SCFAs concentration. In conclusion, this study suggests that VWS holds potential prebiotic properties for human health and provides insights into the potential benefits of grape processing by-products on gastrointestinal and colonic health. It also presents a feasible scientific basis for improving the utilization of VWS in functional foods to increase the use of wine by-products and to reduce environmental pollution caused by their indiscriminate disposal.

## Figures and Tables

**Figure 1 foods-11-02615-f001:**
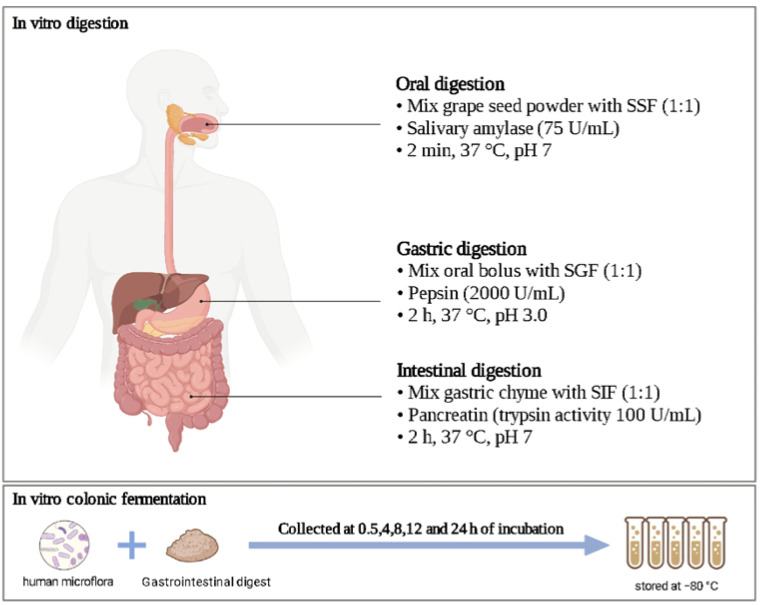
Diagram of VWS digestion and fermentation in vitro.

**Figure 2 foods-11-02615-f002:**
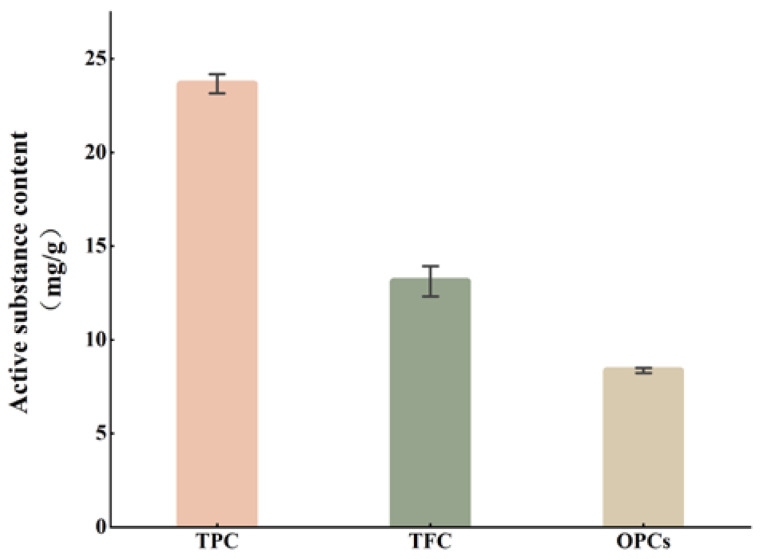
Active substance content of VWS.

**Figure 3 foods-11-02615-f003:**
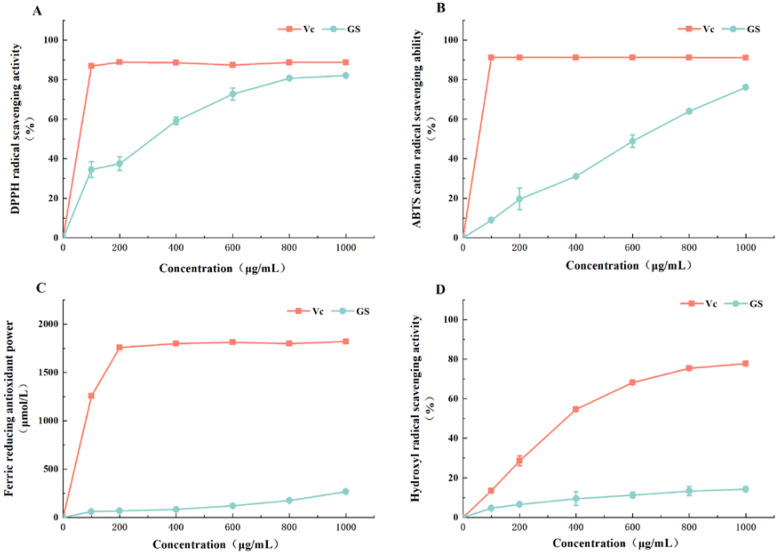
In vitro antioxidant activity of VWS. (**A**) DPPH radical scavenging activity; (**B**) ABTS radical cation scavenging activity; (**C**) Ferric reducing antioxidant power; (**D**) Hydroxyl radical scavenging activity.

**Figure 4 foods-11-02615-f004:**
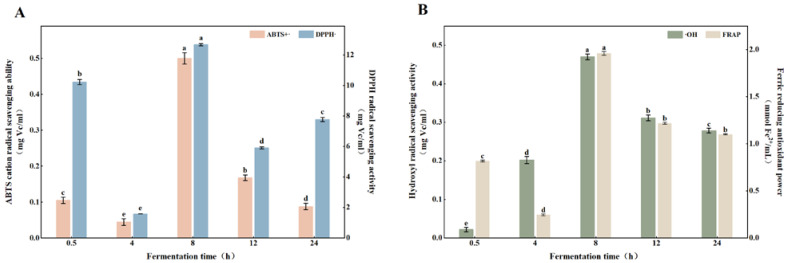
In vitro antioxidant activity of fermentation broth during in vitro fermentation. (**A**) ABTS and DPPH radical scavenging activity; (**B**) Hydroxyl radical scavenging activity and ferric reducing antioxidant power. Means with different letters in figures were significantly different at *p* < 0.05.

**Figure 5 foods-11-02615-f005:**
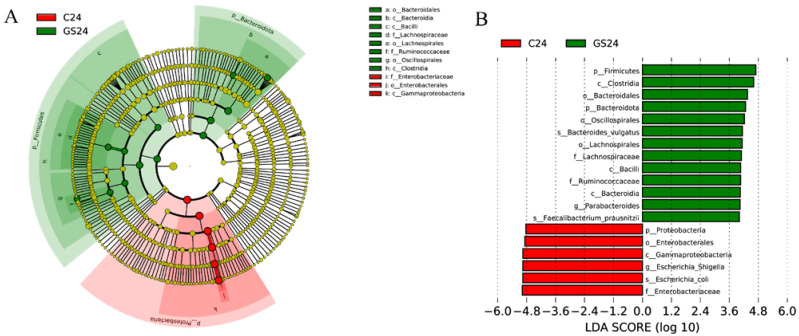
LEfSe analysis (**A**) LEfSe evolutionary branching diagram (red nodes indicate enrichment in C24, while green nodes indicate enrichment in G24) (**B**) Histogram of LDA value distribution (LDA > 4).

**Figure 6 foods-11-02615-f006:**
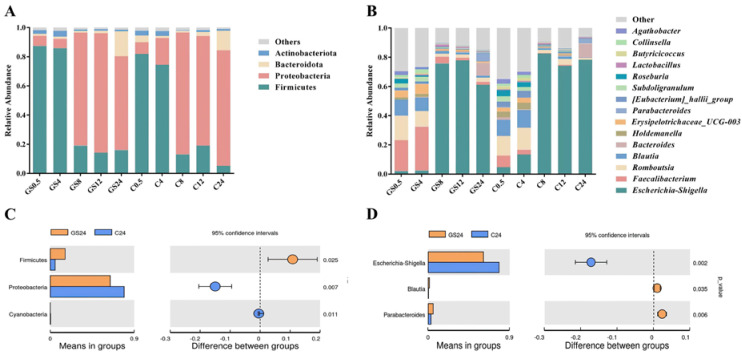
Changes in the relative abundance of intestinal microbiota. (**A**) Species distribution map at the phylum level; (**B**) species distribution map at the genus level; (**C**) analysis of differences in microbial composition at the phylum level for 24 h in vitro fermentation; (**D**) analysis of differences in microbial composition at the genus level for 24 h in vitro fermentation (*p* < 0.05).

**Figure 7 foods-11-02615-f007:**
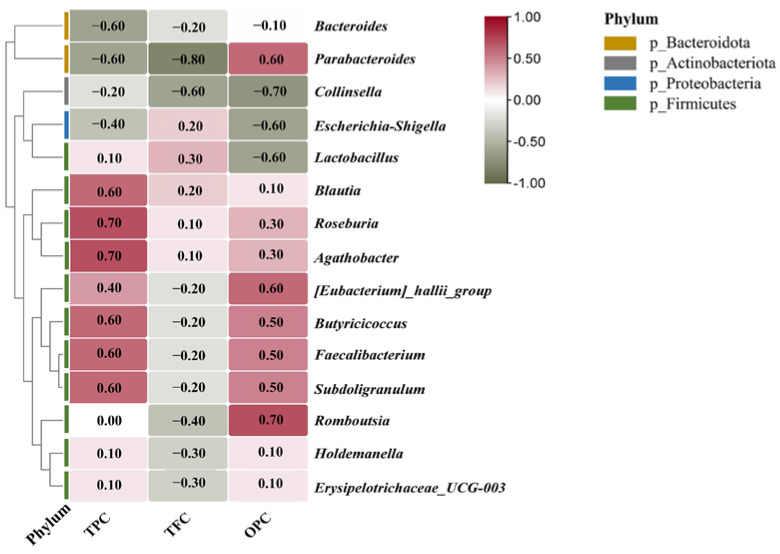
Heatmap of the correlation between TPC, TFC, and OPC in group GS and changes in intestinal micro-organisms during fermentation.

**Figure 8 foods-11-02615-f008:**
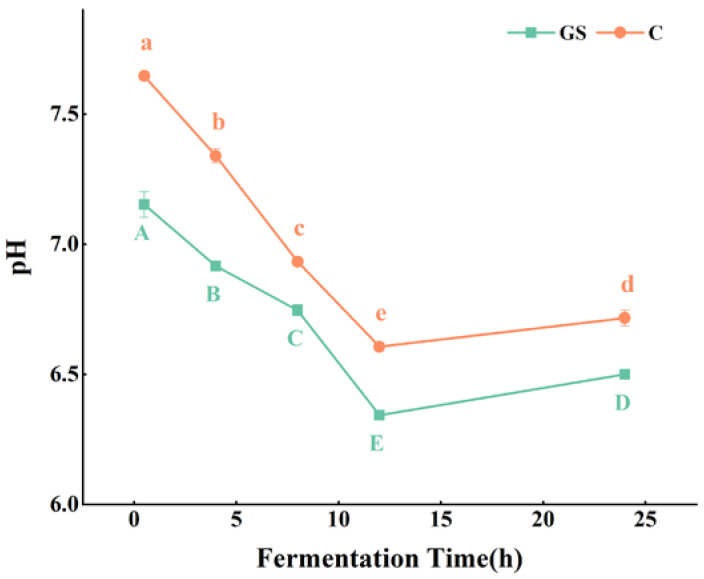
Change of pH during in vitro fermentation. Means with different letters in figures were significantly different at *p* < 0.05.

**Figure 9 foods-11-02615-f009:**
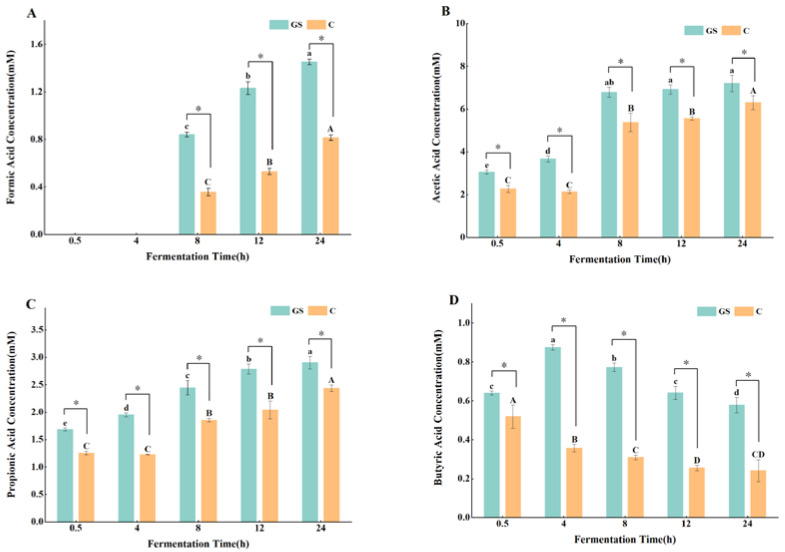
Change of SCFAs concentration during in vitro fermentation. (**A**) Formic acid concentration; (**B**) Acetic acid concentration; (**C**) Propionic acid concentration; (**D**) Butyric acid concentration. Means with different letters in figures were significantly different at *p* < 0.05, * *p* < 0.05.

**Table 1 foods-11-02615-t001:** Changes in the total phenol, total flavonoid, and procyanidin contents during in vitro fermentation.

Fermentation Time/h	TPC μg GAE/mL	TFC μg RE/mL	OPC μg C/mL
0.5	69.44 ± 0.22 ^b^	254.60 ± 1.33 ^b^	19.05 ± 0.39 ^c^
4	47.56 ± 1.74 ^c^	94.81 ± 0.69 ^e^	7.67 ± 0.40 ^e^
8	101.06 ± 0.62 ^a^	363.09 ± 1.23 ^a^	50.58 ± 1.68 ^a^
12	15.31 ± 0.49 ^d^	230.86 ± 1.43 ^c^	37.14 ± 2.16 ^b^
24	13.11 ± 0.31 ^d^	134.20 ± 0.97 ^d^	12.99 ± 1.56 ^d^

Data expressed as mean ± standard deviation (*n* = 3). Different letters in the same column of data indicate significant differences between groups (*p* < 0.05).

**Table 2 foods-11-02615-t002:** Alpha-diversity values of samples.

Sample	OTUs	Chao1	ACE	Shannon	Simpson
C0.5	489	612.645	622.445	5.728	0.958
C4	495	609.328	608.372	5.311	0.938
C8	321	384.683	373.15	1.764	0.318
C12	359	395.212	405.988	2.415	0.448
C24	288	337.752	334.395	1.857	0.384
GS0.5	532	696.779	731.194	4.961	0.912
GS4	569	710.306	745.204	4.782	0.877
GS8	265	301.14	304.607	2.153	0.422
GS12	272	313.288	319.05	2.027	0.39
GS24	303	345.193	346.17	3.035	0.618

## Data Availability

The datasets used and analyzed during the current study are available from the corresponding author on reasonable request.

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
