# Peer review of "Antioxidant Activity of Vitis davidii Foex Seed and Its Effects on Gut Microbiota during Colonic Fermentation after In Vitro Simulated Digestion"

_foods, 2022, doi:10.3390/foods11172615_

Round 1

Reviewer 1 Report

Manuscript submited for review investigates the possibility of utilization of Vitis davidii Foex whole seed, by product of wine industry, via in vitro gastrointestinal digestion and colonic fermentation models, which were used to simulate the digestion and absorption of Vitis davidii Foex whole seed in humans, explore the interaction between Vitis davidii Foex whole seed and intestinal microbiota, and elucidate  the potential effects of the components in Vitis davidii Foex whole seed on intestinal health through their joint action.

The research is conducted in compersensive manner with wholesome approcach in samples’ analysis.

Introduction section is appropirate and references literature is contemporary.

Material and methods section is detailed and allows repetition of performed research,

Results and discussion section is adequatly presented with satisfactory referencing to other sources.

This section presened the results of phenol content and antioxidative activity of Vitis davidii Foex whole seed testing; the effect of in vitro faecal fermentation on the dynamic change of phenol content and antioxidative activity; the effect of Vitis davidii Foex whole seed digest on intestinal microbiota and the effect of Vitis davidii Foex whole seed digest on pH and SCFAs production during in vitro fermentation.

Conclusion section provides usefull and practical conslusions adequatly derived form presented results.

Some minor corrections are noted in manuscript pdf file.

Reviewer 2 Report

Pleasure to read and very well written. Very interesting work.

Reviewer Comments:

1.     The reviewers fail to mention the term prebiotic, which seems to be the premise of the grape pomace on gut microbiota. Previous research of other grape pomace/seeds used as a prebiotic needs to be discussed.

2.     Why were “V. davidii Foex seeds were dried and crushed to 80-mesh powder”. What was the reasoning behind this size. Was the effectiveness of non-crushed seeds that had gone through oral-gastric simulator examined?

3.     Figure 4: Genus and species names need to be larger. Also, the authors discuss microbiota change in genus but also mention phylum comparison as well (Firmicutes). Authors need to make the comparison more understandable and explain which taxonomic level is being compared.

4.     Figure 7: Might consider moving to supplemental information.

5.     Why were the specific hours of fermentation (i.e., 8, 24) chosen?

6.     In the conclusion authors state the treatment led to a reduction of certain gram negative bacteria and increased population of beneficial microbes. How can this result be compared to other prebiotics on the market? Is the significance great enough to justify the use over current dietary prebiotics?

Reviewer 3 Report

The manuscript foods-1872849 entitled “Antioxidant activity of Vitis davidii Foex seed and its effects on gut microbiota during colonic fermentation after invitro simulated digestions” described antioxidant activity, digestibility  and fermentability of a by-product, V. davidii Foex seed under simulated gastrointestestinal conditions. The authors tried to present potential of V. davidii Foex seed on modulation of human gut microbiota. The results showed that during simulated gastrointestinal conditions, VWS released polyphenol, flavonoids, and proanthocyanin. The rest VWS after simulated small intestine condition exhibited partial inhibitory effect against Escherichia-Shigella; however, its fermentability is still unclear (figure 5) as the changes in the relative abundance of intestinal microbiota from 0.5-24 h of both treatment and control were likely similar. The authors tried to compare these changes of treatment with control, but they presented only at 24 h of the fermentation. Some methodologies were not completely described. Moreover, several parts of results were presented unclearly. Therefore, it is difficult to make a decision without clarification. Comments and suggestions are as follows.

1. Line 82: please check correct spelling of “Folin-Ciocalteuf’s”

2. Line 89: Is triangular flask referring to Erlenmeyer flask?

3. Line 90:  please clarify “a ratio of 1:12” does it mean 1:12 (w/v) ?

4. Line 90: I don’t understand “the filtrate was then extracted five times” Is it “the slurry was then extracted five times with the same extraction solution”?

5. Line 91: Please describe how was the filtrate obtained. Which technique?

6. Line 98: Please add (mg GE/g) after “gallic acid equivalents per gram VWS.

7. Line 102: Please add (mg RE/g) after “routine equivalents per gram VWS.

8. Line 108: Please add (mg CE/g) after “catechin equivalents per gram VWS.

9. Section 2.4, please add some more detail about calculating antioxidant activity.

10. Section 2.5.1, I don’t understand the purpose of the experiments. After in vitro digestion of VWS, if I don’t misunderstand, the authors separated the precipitate for colonic fermentation. However, the supernatant was just stored at -20oC without any explanation for next experiment.

11. Section 2.5.2, please provide more information about in vitro fermentation. Fermentation medium as well as the VWS content must be stated.

12. Line 142, change “obtaine” to “obtained”

13. Line 159, please correct “iin accordance”

14. Line 191, please correct “was mixe”

15. Line 192, please change “10,000 r/min” to “10,000 rpm”

16. All figures need well preparation due to font size is too small.

17. Figure 2 need further explanation for figure legend, specifically figure 1A and figure 2B-figure 2E. please also provide figure symbols. What is Vc and GS?

18. Line 228, correct “levelled off”

19. Line 235, please provide full name of Vc for the first time, so it will be easy for readers to understand.

20. Line 244-245, is it possible that antioxidant activity of VWS might be from vitamin C in the extract?

21. Line 247, what is Lv?

22. Table 1 should be presented under the text.

23. What is the 8th h?

24. Line 302, please correct Fe2+/mL and when the authors need to state the abbreviated style for vitamin C, please use only either Vc or VC throughout the manuscript.

25. Figure 3, the author needs to clarify for statistical analysis since the current form is unacceptable. What factors would the authors like to compare in Figure 3A, the ABTS or DPPH activity at different time intervals or both ABTS and DPPH activities at different time intervals? There is no statistical analysis in Figure 3B, please also provide it in acceptable form. Please also consider statistical comparison in figure 7.

26. Line 325, please italicize “Clostridium perfringens”

27. Line 326, “bacteria, increase”, please remove comma (,).

28. Line 363, correct “genus lever”

29. Line 363, please italicize “Parabacteroides”

30. Line 367, there is no explanation in figure legend for figure 4B.

31. I don’t understand the authors. They sometimes added references in their results such as line 347 [44], line 372 [46].

32. The authors did not describe results and discussion that is present in figure 5C and figure 5D.

33. Line 411-413, I would not agree with the authors if without control. This is too bias!! As there is unclear information about fermentation medium used for in vitro study, I could not clearly evaluate the results in this section. There are many factors affecting microbiota, majority fermentation medium and bioactive compounds. In figure 6, the authors need to present heatmap of control in order to confirm that TPC and OPC were negatively correlated with Escherichia-Shigella (line 411-413), and further enhanced beneficial bacteria, Blautia, Roseburia, Butyricicocus, and Faecalibacterium. What kind of active ingredients in VWS did they play important role in changes in the relative abundance of intestinal microbiota?

34. Line 431, Did figure 6 be derived from GS24? Please also include this information in the figure.

35. Line 433, please correct name of section 3.5.

36. Section 3.5, typically, organic acids are from fermentation of carbohydrates. However, in this study, only VWS was applied in the fermentation medium. Could the authors provide information to explain that why organic acids were produced during the fermentation?

36.  Line 501-502, Line 26-30, I don’t agree with the authors since VWS did not show significant modulation of microbiota in contrast it exhibited partial inhibition of Escherichia-Shigella.

37. Please edit abbreviated journal in references 4, 7, 17, 23, 24, 28, 33, 35, 37, 38, 39, 50, 62

Reviewer 4 Report

In general, this work is well-designed, write and discuss. However, figures must be improved in order to understand the information. At this form, it is not possible to publish the manuscript. 

Please separate figure 2 A and B-C-D-E in two independent figures. It is not clear for the reader the presented information of results. The same comment for the rest of figures. Additionally, the resolution of the images must be improved.

Line 227: Remove 80.70%±0.35% by 80.70 ± 0.35%. Revise and modify in all the manuscript.

Round 2

Reviewer 3 Report

Dear authors

The manuscript foods-187849-v1 have been substantially revised according to my comments and suggestions. However, there are still unclear information and misleading statistical analysis, and some methods are unclear to me and further readers. Following is my comments and suggestions.

1. Line 198, please edit “1:12” to 1:12 (w/v)

2. Line 122, please clarify 1000 μg/mL of what substance (initial VWS or lyophilized powder of VWS extract)

3. Line 124, edit 4:1 “ratio to obtain” to 4:1 (v/v) ratio to measure

4. Line 136, add (v/v) after 1:1

5. Line 138, add (v/v) after 1:1

6. Line 140-141, please make the sentence more clearly “The precipitate was lyophilized and stored separately from the supernatant”. What was the lyophilized precipitate used for? and what was the supernatant used for?

7. Line 161, I understand that the digested VWS was in solid form as described in line 141. How did the authors prepare the digested mixture? Please make this sentence more clearly.

8. Line 233, please abbreviated Vitis quinquangularis to “V. quinquangularis”

9. Line 138, please include this information to the figure legend “Different letters indicate significant differences (p<0.05) of antioxidant activity obtained from each antioxidant activity assay”

10. Line 429, please add “Figure 6D” after “that in group C”.

11. Figure 8, according to information in line 466-467, I recommend the authors to dependently compare pH value within control fermentation (5 pH value) and GS fermentation (another 5 pH values). Please also indicate definition of “different letters” in the figure legend”.

12. Once again, figure 9, please consider modifying the multiple comparison of each SCFA within each fermentation (control and GS)

13. Line 509, please remove “pH and” because there is no information about change in pH in figure 9.

14. I found many mistake in references. Please consider editing reference style in the correct format.

Some references contain scientific name which has not been italicized such as references 1, 25, 32, 37, and 52

Some references contain inappropriate journal abbreviation. They contain "full stop (.)" which need to be removed such as references 25, 27, 30, 39-41, and 65.

Some references have been capitalized each word of their title such as references 16, 31, 43, and 59. Please edit them by using only "sentence case" (capitalize only the fist word).

Line 568 please change "Food & Functions" to "Food Funct"

Line  592, please change "Lwt" to "LWT"
